# Disruption of Irisin Dimerization by FDA-Approved Drugs: A Computational Repurposing Approach for the Potential Treatment of Lipodystrophy Syndromes

**DOI:** 10.3390/ijms24087578

**Published:** 2023-04-20

**Authors:** Lorenzo Flori, Simone Brogi, Hajar Sirous, Vincenzo Calderone

**Affiliations:** 1Department of Pharmacy, University of Pisa, Via Bonanno 6, 56126 Pisa, Italy; lorenzo.flori@farm.unipi.it (L.F.); vincenzo.calderone@unipi.it (V.C.); 2Bioinformatics Research Center, School of Pharmacy and Pharmaceutical Sciences, Isfahan University of Medical Sciences, Isfahan 81746-73461, Iran; h_sirous@pharm.mui.ac.ir

**Keywords:** drug repurposing, FDA-approved drugs, computational methods, irisin, lipodystrophy

## Abstract

In this paper, we present the development of a computer-based repurposing approach to identify FDA-approved drugs that are potentially able to interfere with irisin dimerization. It has been established that altered levels of irisin dimers are a pure hallmark of lipodystrophy (LD) syndromes. Accordingly, the identification of compounds capable of slowing down or precluding the irisin dimers’ formation could represent a valuable therapeutic strategy in LD. Combining several computational techniques, we identified five FDA-approved drugs with satisfactory computational scores (iohexol, XP score = −7.70 kcal/mol, SP score = −5.5 kcal/mol, ΔG_bind_ = −61.47 kcal/mol, ΔG_bind_ (average) = −60.71 kcal/mol; paromomycin, XP score = −7.23 kcal/mol, SP score = −6.18 kcal/mol, ΔG_bind_ = −50.14 kcal/mol, ΔG_bind_ (average) = −49.13 kcal/mol; zoledronate, XP score = −6.33 kcal/mol, SP score = −5.53 kcal/mol, ΔG_bind_ = −32.38 kcal/mol, ΔG_bind_ (average) = −29.42 kcal/mol; setmelanotide, XP score = −6.10 kcal/mol, SP score = −7.24 kcal/mol, ΔG_bind_ = −56.87 kcal/mol, ΔG_bind_ (average) = −62.41 kcal/mol; and theophylline, XP score = −5.17 kcal/mol, SP score = −5.55 kcal/mol, ΔG_bind_ = −33.25 kcal/mol, ΔG_bind_ (average) = −35.29 kcal/mol) that are potentially able to disrupt the dimerization of irisin. For this reason, they deserve further investigation to characterize them as irisin disruptors. Remarkably, the identification of drugs targeting this process can offer novel therapeutic opportunities for the treatment of LD. Furthermore, the identified drugs could provide a starting point for a repositioning approach, synthesizing novel analogs with improved efficacy and selectivity against the irisin dimerization process.

## 1. Introduction

Pathologies involving adipose tissue have wide genotypic diversity and are nearly always associated with the presence of other related metabolic imbalances. This complex phenomenon leads to a huge range of phenotypic changes. Among the first aspects to be considered are the amount and distribution of fat mass. On one side of this great gap, there is obesity and more complex levels of metabolic syndrome, presenting as excessive white fat deposits; on the other, there is lipodystrophy (LD), characterized instead by an abnormal loss of white and subcutaneous adipose tissue [1,2].

LDs are a heterogeneous group of genetic diseases characterized by a specific and selective loss of adipose tissue without a reduction in caloric intake, accompanied by muscle hypertrophy [1]. In the most severe and generalized forms, the reduction in adipose tissue can involve the whole body, whereas the partial forms affect only some adipose areas. These changes affect the metabolic aspects, which include wide fluctuations in leptin levels leading to altered hunger/satiety signals from the central nervous system, hyperphagia, and lipid accumulation in several organs (including the liver and skeletal muscle) by inducing a state of insulin resistance [3]. Later, this pathological picture leads to diabetes, hypertriglyceridemia, non-alcoholic fatty liver disease (NAFLD), and polycystic ovary syndrome [4]. In addition to these common features, each type of LD has specific comorbidities, such as cardiovascular complications (e.g., atherosclerosis, arrhythmias, cardiomyopathy), skin problems, skeletal damage, or premature neurological aging [5,6,7]. The mechanism and position of fat loss depend on the biological functions of mutated genes (Table 1) [8]. Diagnosis is performed based on the loss of adipose tissue, body composition analysis, investigation of personal and family history, and molecular, genetic, and imaging tests [6].

In this scenario, irisin represents one of the interesting proteins involved in molecular pathways that are closely associated with the physiological functions of adipose tissue. The mentioned protein is a myokine released by several districts (mainly from the skeletal muscle, heart, and brain) following proteolytic cleavage of the N-terminal extracellular portion of the membrane protein FNDC5. Irisin is involved in the browning process responsible for the phenotypic switching of the white adipose tissue (WAT)—appointed to lipid storage—in beige adipose tissue. Interestingly, beige adipose tissue acquires metabolic features much more similar to those exhibited by the brown adipose tissue (BAT), characterized by reduced lipid depots, multiplicity of intracellular organelles, high mitochondrial expression, and consequent high metabolic activity. Beige adipose tissue no longer has only the physiological function of storing lipids, but also that of consuming them to produce ATP and releasing heat. Muscle hypertrophy causes an increase in irisin plasma levels and triggers the browning process, reducing the WAT and lipid components with the development of beige adipose tissue [9,10].

Although there are more shadows than light with respect to irisin, it was observed that its X-ray crystallography structure consists of an N-terminal FNIII-like domain, which forms a continuous intersubunit β-sheet dimer and binds to a flexible C-terminal tail. This experimental evidence led to the hypothesis that dimerization may be a fundamental mechanism for the irisin receptor’s activation and, therefore, for the triggering of all of the autocrine/paracrine signaling mechanisms in which it is involved [11].

A case–control study involving 51 subjects (19 with familial partial LD with LMNA heterozygous pathogenic variants (FPLD), 13 obese non-diabetic subjects (OND), and 19 healthy normal-weight control subjects (HC)) demonstrated how FPLD subjects have a high intra-abdominal/total abdominal fat ratio compared with the HC and OND groups, reflecting the drastic reduction in subcutaneous adipose tissue in subjects with LD. The irisin level in the FPLD group displayed an increasing trend. The irisin/leptin ratio was significantly enhanced compared with HC and OND (Figure 1) [2]. Consequently, the disruption of irisin dimers can be pursued as an appealing strategy to counteract the overactivation of irisin function in subjects with LD.

In addition, recent studies have shown that some amino acids on the surface of the irisin monomers specifically play vital roles in the formation of the dimer. In this context, R72 from C′-loop; L74, R75, F76, Q78, E79, and V80 from C′-strand; A88 and W90 from E-strand; and D91 from E-loop are the most important residues involved in the direct interaction between the two chains (Figure 2) [11,12].

With a view to new therapeutic interventions that are effective in the treatment of rare pathologies for which therapeutic options are poor or missing, the repurposing/repositioning of approved drugs represents a viable and interesting strategy. Drug repurposing is a strategy for identifying new uses for approved or investigational drugs that fall outside the scope of the approved therapeutic indication [13,14]. Redirecting a drug to a different therapeutic indication has a higher likelihood of success and significantly reduced time and development costs compared to developing a new drug. There are many scientific approaches to drug repurposing. From this point of view, an innovative and effective approach is represented by computational repositioning based on molecular modeling techniques such as ligand- and structure-based methodologies. Ligand-based methods rely on the concept that similar molecules tend to have similar pharmacological profiles. In drug repurposing, ligand-based techniques have been extensively employed for analyzing and predicting the activity of compounds against novel drug targets. Today, ligand-based methods take advantage of the increasing availability of a lot of public biological/pharmacological/structural data and the improvement in computational techniques (e.g., machine learning methods) [15,16]. Considering the structure-based approaches, molecular docking allows the investigation of how a drug (ligand) interacts with a protein or protein complex, while molecular dynamics (MD) simulations can be used to examine thousands of frames, both to investigate changes in drug–protein interactions and to understand their role in cellular mechanisms [17,18]. In the last decade, computational approaches have guided interventions for drug repurposing; many examples are available in the literature, where novel possible therapeutic indications have been found for existing drugs [19,20,21,22,23,24,25], including the possibility of drug repurposing in the field of rare diseases [26,27]. Again, during the last pandemic global health scenario, computer-based drug repurposing approaches were exploited to provide drugs to be repurposed against SARS-CoV-2, so as to obtain fast therapeutic interventions [28,29,30,31,32,33,34].

Taken together, this information can be significant to set up a computational repurposing protocol based on molecular docking, along with the evaluation of ligand binding energy coupled with MD simulations, aimed at identifying potential drugs that are able to interact with the surface of the irisin monomer, precluding the correct interaction for forming the irisin dimer. To this end, for the first time, we explored the possibility of finding FDA-approved drugs that are capable of interfering with irisin dimer formation. These drugs can be considered as appropriate starting points for further explorations as possible therapeutic options to treat LD, in which irisin dimerization is pathologically relevant. 

## 2. Results and Discussion

### 2.1. High-Throughput Docking

After the appropriate preparation of the irisin monomer and the approved/investigational drugs database, for a total of 6901 drugs (further details are reported in the Materials and Methods section), we started the high-throughput docking campaign considering two scoring functions available in Glide software: (Glide, Schrödinger, LLC, New York, NY, USA, 2020) the extra precision (XP) and the standard precision (SP). After the ranking of compounds for each scoring function, for a total of 237 entries, the results were merged, considering approximately the top 1% of compounds of the docked solution. As a result, only compounds that were able to strongly interact with the key residues involved in the dimerization (visual inspection) with the unique binding mode were finally selected, considering both scoring functions (XP and SP), docking scores lower than −5.00 kcal/mol, and a satisfactory ΔG_bind_ (Table 2).

Accordingly, the drugs that met the filtering criteria were considered for further in silico investigation. In particular, the complexes obtained from molecular docking studies were employed as starting points for the MD simulation experiments in order to evaluate their binding stability. The results of the most promising FDA-approved drugs in establishing and maintaining significant interactions with the irisin interface involved in dimerization are presented in detail in the next paragraphs.

### 2.2. Molecular Dynamics Simulation

In order to increase the reliability and the robustness of the ligand screening, an extensive MD investigation was conducted using Desmond software, (Desmond Molecular Dynamics System, D. E. Shaw Research, New York, NY, USA, 2020. Maestro-Desmond Interoperability Tools, Schrödinger, New York, NY, USA, 2020) considering 23 complexes obtained from molecular docking studies. MD simulation experiments are crucial in this kind of protocol, considering that the binding site is located on the surface of the protein and, thus, is of pivotal importance to assess whether a selected drug is able to maintain the contacts with the selected binding site on the irisin monomer, or whether it might break away from the binding site during the protein dynamics. Accordingly, all of the trajectories of MD runs were studied, in order to select only those drugs that were able to form productive interactions with the irisin monomers while preserving the binding with the considered protein subunit (Table 2). All of the trajectories were analyzed, calculating the root-mean-square deviation (RMSD) and root-mean-square fluctuation (RMSF), and evaluating the dynamic ligand-interaction diagrams, combined with the visual inspection of each trajectory. All of the calculations regarding the MD investigation are reported in the Appendix A and labelled as Appendix A. In general, the MD simulation output showed a small RMSD for the irisin monomer along with limited fluctuations, as indicated by the RMSF. 

The most promising FDA-approved drugs that could be involved in disrupting irisin dimerization are presented in detail in the main text of this article in the following section.

### 2.3. Potential Hit Compounds Disrupting Irisin Dimerization

#### 2.3.1. ZINC000003830946 (Iohexol)

One of the most promising FDA-approved drugs selected by the mentioned computational protocol is represented by ZINC000003830946 (iohexol). Figure 3 shows the molecular docking output, highlighting the main contacts that this drug can establish with the surface of irisin dimers (Figure 3A). In particular, this compound was able to strongly target the irisin interface, establishing H bonds with R75, E79, T82, T84, S86, W90, and D91 (Figure 3B). As visible in the pictorial representation, this compound was able to occupy the whole surface involved in the irisin dimerization, and this could be helpful for disrupting the correct recognition of irisin monomers.

As mentioned for improving the quality of the screening, we evaluated the binding stability by conducting MD simulation experiments in explicit solvents. The outputs, in terms of the RMSD and RMSF of ZINC000003830946, are reported in Appendix A, while the trajectory analysis in terms of binding stability is presented in Figure 4. The compound ZINC000003830946 was able to maintain the binding with the irisin binding site during the MD simulation, maintaining the main contacts found in the molecular docking studies. In particular, the H bonds with residues E79, T82, T84, S86, A88, W90, and D91 were strictly maintained, while the contact with R75 became less evident, and a more favorable H bond with Q66 was detected, established with the same moiety of the molecule. The outputs confirmed the ability of ZINC000003830946 to establish fruitful interactions with the irisin monomer surface involved in the dimerization process.

#### 2.3.2. ZINC000060183170 (Paromomycin)

Another interesting FDA-approved drug selected by the mentioned computational protocol is represented by ZINC000060183170 (paromomycin). Figure 5 illustrates the molecular docking outputs, highlighting the main contacts that this drug can establish with the surface of the irisin monomer (Figure 5A). In particular, the considered compound can strongly target the key residues involved in the dimerization process and, in particular, could form H bonds with R75, Q78, S86, and W90 (Figure 5B). Furthermore, we observed an important network of polar contacts with E79 (H bonds and salt bridges). Moreover, in this case, the mentioned drug fitted nicely with the surface of the irisin monomer, possibly impeding the correct interactions between irisin monomers.

To validate the docking outputs, we examined the MD simulation trajectories to observe whether the compound showed the capability to preserve the binding mode found in the molecular docking studies. The output of this analysis is reported in Figure 6, while the calculation of RMSD and RMSF for this complex is reported in Appendix A. In particular, we observed that the strong network of polar contacts with E79 was neatly retained, as was the H-bond interaction with Q78. The contact with R75 was found to be mainly water-mediated, while the contact with S86 became sporadic. Moreover, in addition to the interaction with W90, which was detectable for the whole simulation, we noted the formation of favorable interactions with the backbone of A88 and with the sidechain of D91 (water-mediated H bonds). Accordingly, due to the stability of the binding within the irisin binding site, it is plausible that this compound getting in the way between the irisin monomers could preclude their recognition for forming the dimer, limiting the activation of the irisin pathway.

#### 2.3.3. ZINC000003803652 (Zoledronate)

The third interesting drug selected by virtual screening is represented by ZINC000003803652 (zoledronate). This compound was able to target a specific region of the irisin binding site formed by R75, W90, and D91. In particular, ZINC000003803652 can target R75 by an H bond and two salt bridges with the phosphate group, and by a cation–π stacking with the aromatic portion of the drug. Moreover, ZINC000003803652 can establish an H bond with the sidechain of D91, and it could be able to target W90 via an H bond and a π–π stacking. The molecular docking output of ZINC000003803652 within the selected binding site is illustrated in Figure 7 (panels A and B). 

The MD simulation analysis regarding ZINC000003803652 is reported in Figure 8 and Appendix A. Considering the in-depth analysis of the whole MD trajectory, we observed that the main contacts found by molecular docking were perfectly conserved throughout the whole simulation. In fact, the compound was able to maintain the strong contacts with R75, W90, and D91. Interestingly, the drug ZINC000003803652 could be able to establish additional contacts with R72, L74, and F76 that could contribute to stabilizing the retrieved binding mode, making this compound perfectly anchored to the binding site and highlighting its potential as a disruptor of irisin dimer formation. 

#### 2.3.4. PID: 11993702 (Setmelanotide)

The molecular docking output of the drug with PID 11993702 (setmelanotide) is depicted in Figure 9. Considering the huge size of this drug, it could be able to target several residues of the irisin monomer’s surface (Figure 9A). In fact, through a series of polar and hydrophobic contacts, it could target residues R75 (cation–π stacking), F76 (backbone, H bond), Q78 (H bond), E79 (H bonds, and salt bridges), S86 (H bond), Q103 (H bond), and Q110 (H bond) (Figure 9B). This binding mode allowed the compound to occupy almost the entire surface of the irisin monomer involved in the dimerization process. 

The MD simulation experiments confirmed the docking outputs. The binding mode was quite stable throughout the whole MD simulation, and the key residues were constantly targeted by the drug. Although we noted small changes in the conformation of the drug (as suggested by the calculation of RMSD), we observed a slight movement of the compound toward the center region of the binding site. In fact, the contacts with residues R75, F76, Q78, and E79 were maintained, while novel, more favorable contacts were established with V80, N81, A88, W90, and D91; obviously, the H bonds with S86, Q103, and Q110 became sporadic and were no longer detectable (Figure 10 and Appendix A).

#### 2.3.5. ZINC000018043251 (Theophylline)

As the last drug with a favorable computational outcome, we identified the compound ZINC000018043251 (theophylline). The molecular docking output is reported in Figure 11. Similarly to the drug zoledronate, we observed a preferred binding to the same region of the surface of the irisin monomer (Figure 11A). The compound ZINC000018043251 was able to target the backbone of A88 via a H bond and could strongly target W90 via a H bond and a double π–π stacking (Figure 11B).

The MD simulation trajectory analysis for the compound ZINC000018043251 highlighted strong stability in targeting the retrieved binding site. In fact, the drug maintained the ability to target A88 and W90, while also showing the formation of a water-mediated H bond with the residue D91, and important contacts with R75 (cation–π stacking), I77 (hydrophobic interactions), and sporadically with E79 (water-mediated H bond) that could contribute to stabilizing the retrieved binding mode (Figure 12 and Appendix A).

Recently, updated methods have considered a combination of molecular mechanical force fields with continuum solvation models, such as MM-GBSA. This method enables us to make more accurate predictions of the binding free energy of ligands from MD trajectories. Considering that all calculations in MM-GBSA are based on the initial and final states of MD simulations, this method has recently emerged as one of the “end-point” approaches in drug discovery protocols. This approach can predict the contributions of a range of important interactions, including both polar and non-polar types, in the protein–ligand binding event. Thus, in the last step of our computational workflow, the trajectory models obtained from the previous MD simulations were subjected to MM-GBSA calculations to further substantiate the binding affinity of the top-ranked hits with favorable thermodynamics towards the target binding site. Accordingly, we used the whole trajectories to predict the ΔG_bind_ of the five promising drugs selected, employing the presented computational protocol. The outputs of this calculation, along with their contributions to the total binding free energy from various energy components, are provided in Table 3. The general inspection of the free energy components in Table 3 revealed that the van der Waals interaction energy (ΔG_vdW_) was the most important contributor to the ligands’ binding energy. This observation emphasizes the critical importance of hydrophobic interactions in the stability of the ligand–protein complexes, which is logical considering the hydrophobic nature of the irisin interface involved in the dimerization process. Furthermore, the calculated ΔG_bind_ throughout the whole MD trajectories for the five drugs showed slight differences with the ΔG_bind_ calculated from the docking poses, highlighting the possibility that the selected drugs could strongly bind to the irisin interface with relevant stability and, consequently, could act as irisin dimerization disruptors. 

### 2.4. Influence of the Most Promising Drugs against the Irisin Dimerization

In order to evaluate the influence of the selected drugs on the assembly of irisin monomers, we evaluated the binding energy of the irisin monomers to form the dimers and how the presence of the selected drugs could influence the irisin dimerization in terms of computational scores. For this purpose, we used HADDOCK 2.4, since this software can evaluate the protein–protein binding energy to form the irisin dimer, considering the irisin monomers alone, and it can consider the influence of small molecules bound to the protein monomer in the formation of the dimer. The results of this investigation are reported in Table 4 and Figure 13 (panels A and B). As easily assessable, and as expected, based on the previous computational data, all of the selected drugs could negatively influence the irisin dimerization. In particular, we detected only three clusters for the irisin monomers alone, with small differences in the binding energies and binding modes. Conversely, when a selected drug was considered in the calculation, taking into account the binding mode found by molecular docking calculations, we noted an increase in the uncertainty of the binding mode, retrieving several clusters related to the possible assembly of irisin. Furthermore, the dimer formation energies were increased when a drug was present, indicating a decrease in the affinity of irisin monomers, reflecting the difficult recognition of the monomers. Accordingly, the mentioned data indicated that the selected drugs could preclude the correct recognition of irisin monomers, physically hindering the irisin dimerization. Remarkably, the selected drugs were able to reduce the affinity of irisin for its counterpart by at least 50%, suggesting that the selected FDA-approved drugs could negatively influence the irisin dimerization. Accordingly, they deserve further investigation in this sense. 

To give further information about the retrieved FDA-approved drugs potentially indicated for a repurposing for treating LD in which there is an increase in irisin dimer formation, we collected marketed information for those drugs, and the analysis is reported in Table 5.

Finally, among the screened compounds, we identified some investigational or approved drugs from different regulatory agencies than the FDA, and we would like to highlight that some of them could be investigated as possible therapeutic options for LD. For example, among the investigational drugs, diquafosol (an agonist of the P2Y2 purinogenic receptor, indicated in dry eye), voglibose (an inhibitor of α-glucosidase with anti-hyperglycemic activity), and salvianolic acid B (an antioxidant agent currently being investigated for the treatment of vascular dementia). Therefore, these compounds may also deserve further investigation in this context, considering that they showed satisfactory computational results comparable to those presented for the FDA-approved drugs.

## 3. Materials and Methods

### 3.1. Computational Details

Calculations were performed on a system comprising 56 Intel Xeon E5-2660 v4@2.00 GHz processors and 2 NVIDIA GeForce RTX 2070 8 GB GPUs, running the Maestro Molecular Modeling environment release 2020-3 (Schrödinger, LLC, New York, NY, USA, 2020).

#### 3.1.1. Protein and World-Approved Drugs Database Preparation

The world-approved drugs dataset was taken from the ZINC database, including FDA-approved drugs (https://zinc.docking.org/; accessed on 10 December 2022) (~4000 drugs) and investigational/experimental drugs (~2500 drugs). Furthermore, we downloaded the FDA-approved drugs dataset from Selleckchem (https://www.selleckchem.com/screening/fda-approved-drug-library.html; accessed on 6 January 2023), since it was updated in January 2023. After removing the redundant structures, we obtained a total of 6901 drugs. The drugs were prepared by means of MacroModel [35] and LigPrep [36], as described in [37,38,39]. In particular, all of the drugs were minimized using MacroModel, employing the force field OPLS3 [40]. A GB/SA solvation model for simulating the solvent effect was used with ‘‘no cutoff’’ for non-bonded interactions. The PRCG method (5000 maximum iterations and 0.001 gradient convergence threshold) was employed. Furthermore, the compounds were then submitted to the LigPrep program, generating possible ionization states at pH 7.4 ± 0.2.

The irisin dimer (PDB ID 4LSD) [11] was downloaded from the Protein Data Bank (PDB) and prepared by means of the Protein Preparation Wizard implemented in Maestro Suite [41], as previously reported [39,42]. The materials used for the crystallization process were removed. Furthermore, in order to accomplish the in silico studies, we considered chain A for the molecular docking calculations.

#### 3.1.2. High-Throughput Docking and Ligand Binding Energy Evaluation

Glide (Grid-Based Ligand Docking with Energetics) was employed for the high-throughput docking procedure using the drugs database and the protein was prepared as described above, applying the Glide extra precision (XP) and standard precision (SP) methods [43]. Energy grids were prepared using the default values of the protein atom scaling factor (1.0 Å) within a cubic box centered on the residues of the selected chain involved in the dimerization process [11,12]. After that, the database containing the approved and investigational drugs was docked into the selected binding site with default parameters. The number of poses entered to post-docking minimization was set to 50. The Glide XP and SP scores were evaluated. The interactions of the drugs with the protein were assessed using the Ligand-Interaction Diagram application available in the Maestro suite. Furthermore, the screening protocol was implemented with the calculation of ligand binding energy performed using Prime software (Prime, Schrödinger, LLC, New York, NY, USA, 2020) [44], as previously described [45,46,47]. In order to achieve further accuracy for our computational protocol, the calculation of relative ligand binding energy (ΔG_bind_) between the ligand and receptor offers a worthwhile post-scoring approach for prioritizing the screened hits with a lower ΔG_bind_. The MM-GBSA approach combines the molecular mechanical (MM) energies with a continuum solvent generalized Born (GB) model for polar solvation, as well as a solvent-accessible surface area (SASA) for non-polar solvation terms. Accordingly, the representative docked poses obtained through the molecular docking studies were submitted to a Prime protocol to obtain the final rankings of the drugs against the selected enzymes. The final selection, considering the top 1% of the compounds present in the database, was performed by combining visual inspection (interaction of drugs with key residues involved in the dimerization) and docking scores (we selected compounds with XP and SP scores < −5.00 kcal/mol, with similar binding modes derived from the different scoring functions), coupled with a satisfactory ΔG_bind_.

The thermal MM-GBSA script available in Desmond (thermal_mmgbsa.py) [48] was used to evaluate the ΔG_bind_ for the selected complexes. This tool used the Desmond MD trajectory, splitting it into individual frame snapshots, and ran each one through MM-GBSA analysis. During the MM-GBSA calculation, 1000 snapshots from the 100 ns MD simulation were used as inputs to compute the average binding free energy. The evaluated ΔG_bind_ scores are reported as average values in the Results and Discussion section, along with the energy components used in the calculation.

#### 3.1.3. Molecular Dynamics

MD simulations were carried out using the Desmond 6.4 academic version, providing by D. E. Shaw Research (“DESRES”), using Maestro 12.6 as graphical interface (Desmond Molecular Dynamics System, D. E. Shaw Research, New York, NY, USA, 2020. Maestro-Desmond Interoperability Tools, Schrödinger, New York, NY, USA, 2020). MD was performed using the Compute Unified Device Architecture (CUDA) API [49] on two NVIDIA GPUs. The 23 complexes derived from the docking studies were imported in Maestro and solvated by the Desmond system builder into an orthorhombic box filled with water, simulated by the TIP3P model [50,51,52,53]. The OPLS force field [40] was utilized for the MD calculations. Na^+^ and Cl^−^ ions were added to provide a final salt concentration of 0.15 M to simulate the physiological concentration of monovalent ions. Constant temperature (300 K) and pressure (1.01325 bar) were employed with NPT (constant number of particles, pressure, and temperature) as an ensemble class. The RESPA integrator [54] was applied to integrate the equations of motion, with an inner timestep of 2.0 fs for bonded and non-bonded interactions within the short-range cutoff. Nose–Hoover thermostats [55] were employed to maintain the constant simulation temperature, and the Martyna–Tobias–Klein method [56] was utilized to control the pressure. Long-range electrostatic interactions were estimated by the particle-mesh Ewald (PME) technique [57]. The cutoff for van der Waals and short-range electrostatic interactions was set at 9.0 Å. The equilibration of the system was performed with the default protocol provided in Desmond, which consists of a series of restrained minimizations and MD simulations applied to slowly relax the system. Consequently, one individual trajectory for each complex of 100 ns was calculated. The MD simulation experiments were repeated twice for each complex to improve the presented results. The trajectory files were analyzed by simulation event analysis and simulation interaction diagram tools implemented in the Maestro graphical interface. The same applications were used to generate all plots concerning the MD simulation experiments presented in this work. Accordingly, the RMSD was evaluated using Equation (1):(1)$${\mathrm{RMSD}}_{x}=\sqrt{\frac{1}{N}\sum_{i=1}^{N}{\left({r^{\prime }}_{i}\left(t_{x}\right)-r_{i}\;\left(t_{\mathrm{ref}}\right)\right)}^{2}}$$
where the RMSD_x_ refers to the calculation for a frame x, N is the number of atoms in the atom selection; t_ref_ is the reference time (typically, the first frame is used as the reference, and it is regarded as time t = 0), and r’ is the position of the selected atoms in frame x after superimposing on the reference frame, where frame x is recorded at time t_x_. This procedure was repeated for every frame in the simulation trajectory. Regarding the RMSF, Equation (2) was used for the calculation:(2)$${\mathrm{RMSF}}_{i}=\sqrt{\frac{1}{T}\sum_{t=1}^{T}<{\left({r^{\prime }}_{i}\left(t\right)-r_{i}\;\left(t_{\mathrm{ref}}\right)\right)}^{2}>}$$
where RMSF_i_ refers to a generic residue i, T is the trajectory time over which the RMSF is calculated, t_ref_ is the reference time, r_i_ is the position of residue i, r’ is the position of atoms in residue i after superposition on the reference, and the angle brackets indicate that the average of the square distance is taken over the selection of atoms in the residue. 

#### 3.1.4. Protein–Protein Docking

To assess the influence of the selected drugs on the binding affinity of irisin monomers, we conducted protein–protein docking using the HADDOCK webserver [58,59] (https://wenmr.science.uu.nl/haddock2.4/; accessed on 13 February 2023) in the absence and presence of the selected drugs [60]. The HADDOCK webserver (High Ambiguity Driven protein–protein DOCKing) uses ab initio docking methods to encode information from identified or predicted protein interfaces in ambiguous interaction restraints (AIRs) to conduct the docking process. In this kind of investigation, regarding the input step, we indicated the interacting residues of each irisin monomer [12] (Figure 2) to guide the docking calculation.

## 4. Conclusions

In summary, we have developed a computational protocol to identify FDA-approved drugs that can interfere with irisin dimerization and, consequently, could be useful as novel therapeutic options in LD. Notably, this kind of computational screening was carried out for the first time in this paper, providing relevant hints as to the possible role of chemical entities in disrupting irisin dimerization. Based on the in silico experiments, we identified five FDA-approved drugs (iohexol, paromomycin, zoledronate, setmelanotide, and theophylline) that are potentially able to strongly interact with the surface of the irisin monomer involved in the dimerization process, affecting the correct recognition between irisin monomers and precluding their effective function. Considering that it has been observed that the concentration of irisin dimers is higher in LD patients than in healthy subjects, it has been suggested that a strategy for reducing the function of this protein via disrupting its dimerization should represent a valuable option for slowing down the progression of the disorder. Accordingly, the identification of FDA-approved drugs that are able to preclude the irisin dimerization could offer a novel opportunity to quickly identify drugs that are useful for the treatment of LD, or as starting point for a repositioning approach, since they can be useful to synthesize novel analogs with improved efficacy and selectivity against the irisin dimerization process. 

## Figures and Tables

**Figure 1 ijms-24-07578-f001:**
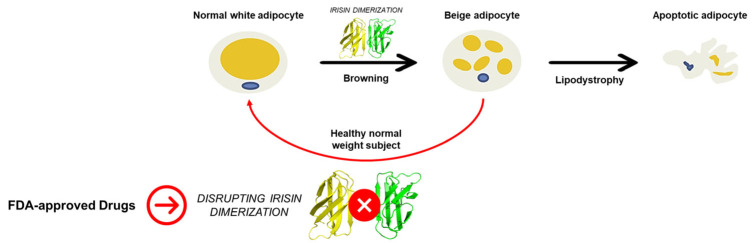
Schematic representation of the relevance of disrupting irisin dimerization in lipodystrophy.

**Figure 2 ijms-24-07578-f002:**
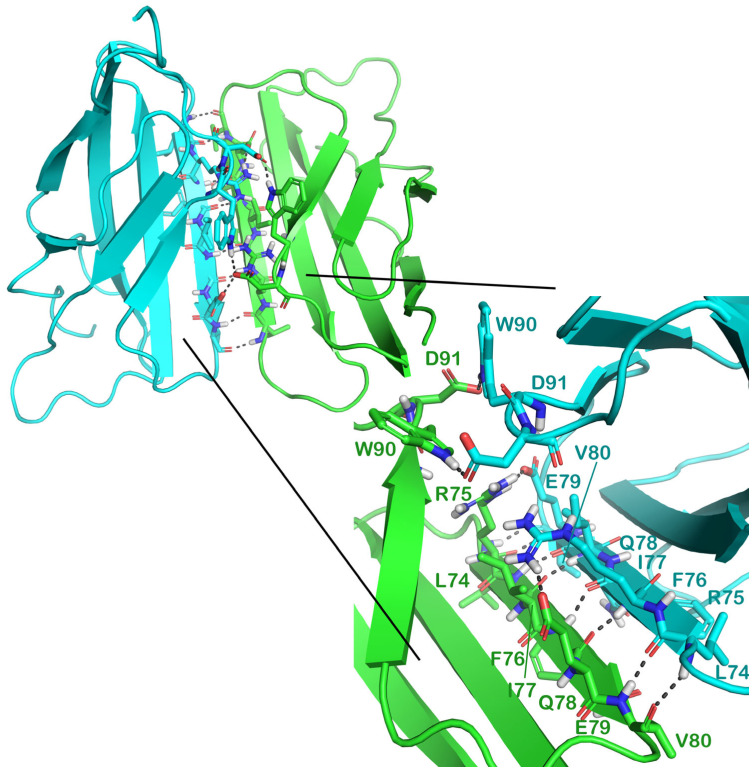
Representation of the irisin dimer, highlighting the interacting surface and related key amino acids: The irisin dimer is represented as a cartoon (PDB ID 4LSD), and each monomer is differently colored, while the interacting amino acids are represented by sticks and the H bonds are indicated as grey dotted lines. The image was generated using PyMOL (the PyMOL Molecular Graphics System, v1.8; Schrödinger, LLC, New York, NY, USA, 2015).

**Figure 3 ijms-24-07578-f003:**
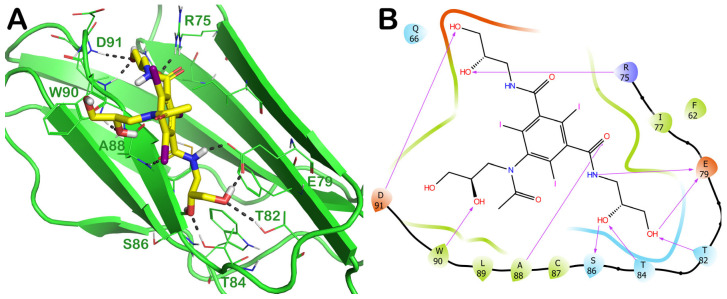
(**A**) Docking output of compound ZINC000003830946 (yellow sticks) within the surface of the irisin monomer (PDB ID 4LSD, green cartoon) involved in the dimerization process. The key interacting residues are represented by lines. The H bonds are depicted as dotted lines. Non-polar hydrogen atoms were removed for the sake of clarity. (**B**) A 2D representation of the contacts established by ZINC000003830946 within the irisin monomer interface. Images were prepared using PyMOL and the Ligand-Interaction Diagram application implemented in Maestro.

**Figure 4 ijms-24-07578-f004:**
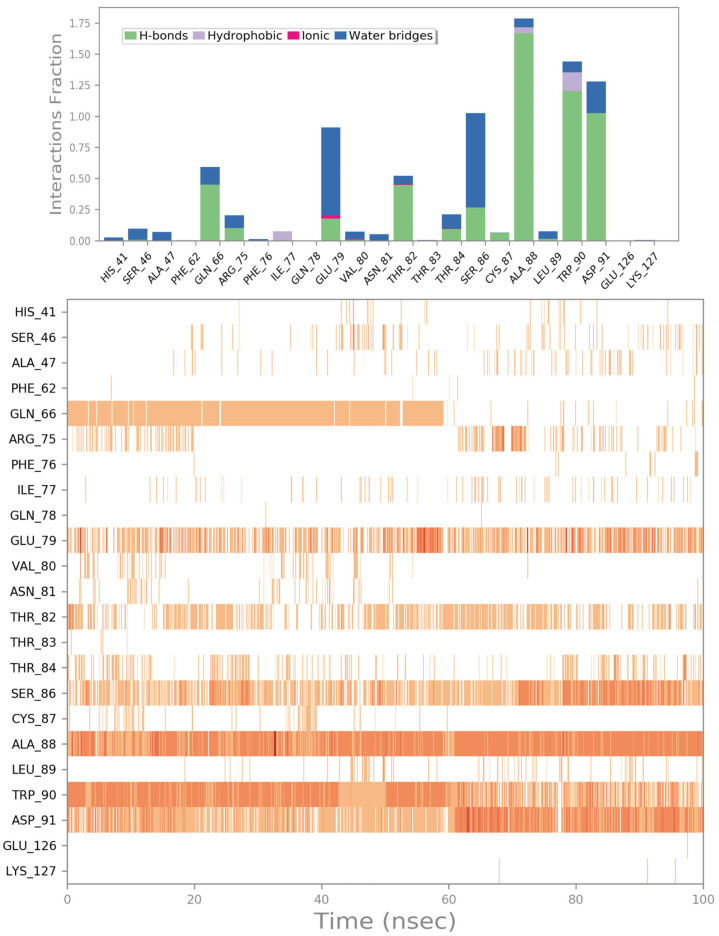
ZINC000003830946 monitored during the MD run. The interactions can be grouped into four types: H bonds (green), hydrophobic (grey), ionic (magenta), and water bridges (blue). The subsequent diagram in the figure illustrates a timeline description of the main interactions. A darker hue of orange indicates that some residues make many distinct contacts with the ligand (Maestro, Schrödinger LLC, release 2020-3).

**Figure 5 ijms-24-07578-f005:**
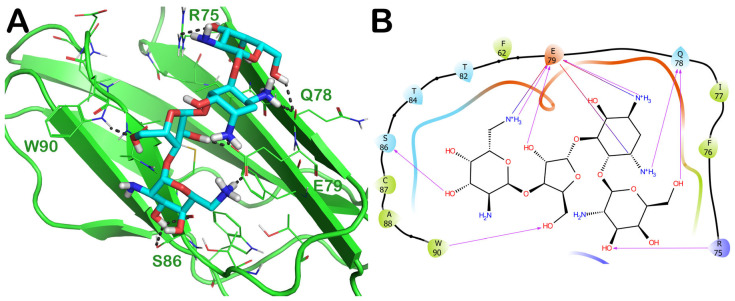
(**A**) Docking output of compound ZINC000060183170 (cyan sticks) within the surface of the irisin monomer (PDB ID 4LSD, green cartoon) involved in the dimerization process. The key interacting residues are represented by lines. The H bonds are depicted as dotted lines. Non-polar hydrogen atoms were removed for the sake of clarity. (**B**) A 2D representation of the contacts established by ZINC000060183170 within the irisin monomer interface. Images were prepared using PyMOL and the Ligand-Interaction Diagram application implemented in Maestro.

**Figure 6 ijms-24-07578-f006:**
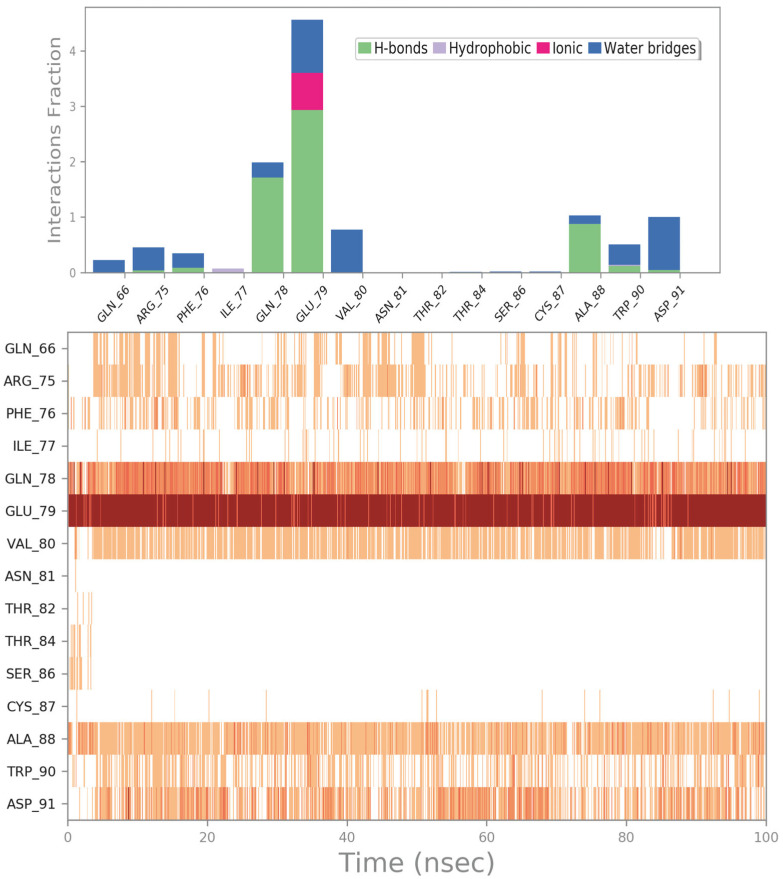
ZINC000060183170 monitored during the MD run. The interactions can be grouped into four types: H bonds (green), hydrophobic (grey), ionic (magenta), and water bridges (blue). The subsequent diagram of the figure illustrates a timeline description of the main interactions. A darker hue of orange indicates that some residues make many distinct contacts with the ligand (Maestro, Schrödinger LLC, release 2020-3).

**Figure 7 ijms-24-07578-f007:**
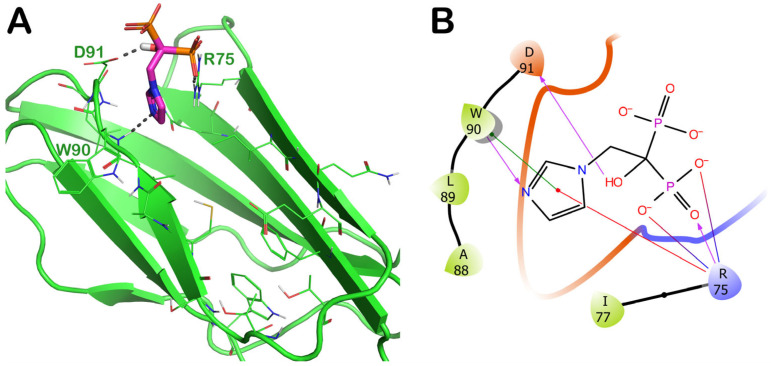
(**A**) Docking output of compound ZINC000003803652 (magenta sticks) within the surface of the irisin monomer (PDB ID 4LSD, green cartoon) involved in the dimerization process. The key interacting residues are represented by lines. The H bonds are depicted as dotted lines. Non-polar hydrogen atoms were removed for the sake of clarity. (**B**) A 2D representation of the contacts established by ZINC000003803652 within the irisin monomer interface. Images were prepared using PyMOL and the Ligand-Interaction Diagram application implemented in Maestro.

**Figure 8 ijms-24-07578-f008:**
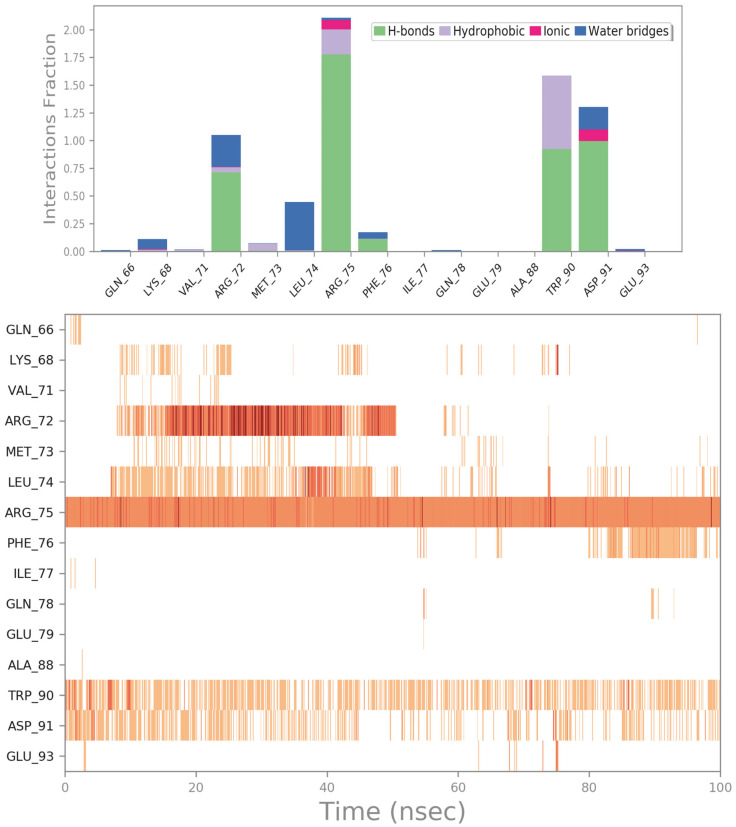
ZINC000003803652 monitored during the MD run. The interactions can be grouped into four types: H bonds (green), hydrophobic (grey), ionic (magenta), and water bridges (blue). The subsequent diagram of the figure illustrates a timeline description of the main interactions. A darker hue of orange indicates that some residues make many distinct contacts with the ligand (Maestro, Schrödinger LLC, release 2020-3).

**Figure 9 ijms-24-07578-f009:**
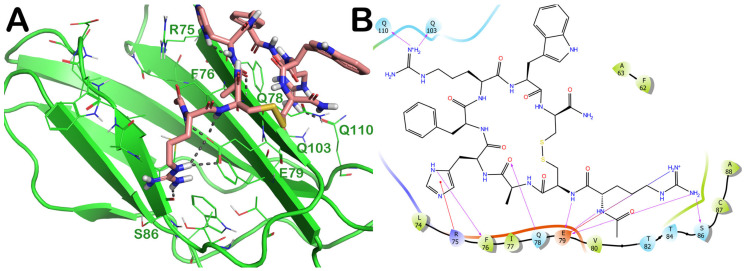
(**A**) Docking output of the drug PID 11993702 (setmelanotide) (light pink sticks) within the surface of the irisin monomer (PDB ID 4LSD, green cartoon) involved in the dimerization process. The key interacting residues are represented by lines. The H bonds are depicted as dotted lines. Non-polar hydrogen atoms were removed for the sake of clarity. (**B**) A 2D representation of the contacts established by compound PID 11993702 within the irisin monomer interface. Images were prepared using PyMOL and the Ligand-Interaction Diagram application implemented in Maestro.

**Figure 10 ijms-24-07578-f010:**
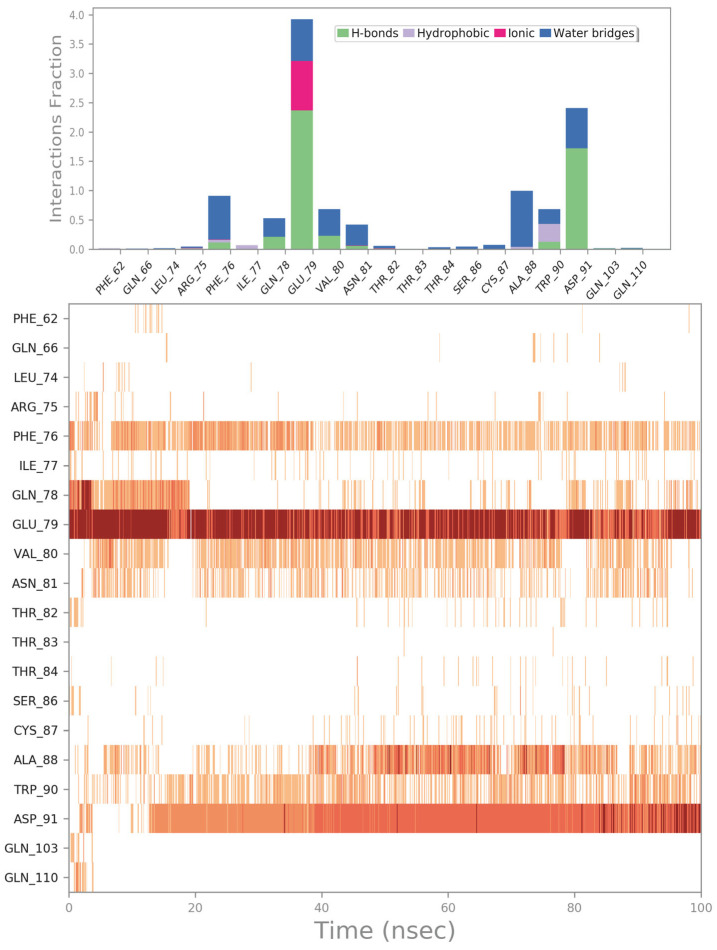
Setmelanotide monitored during the MD run. The interactions can be grouped into four types: H bonds (green), hydrophobic (grey), ionic (magenta), and water bridges (blue). The subsequent diagram of the figure illustrates a timeline description of the main interactions. A darker hue of orange indicates that some residues make many distinct contacts with the ligand (Maestro, Schrödinger LLC, release 2020-3).

**Figure 11 ijms-24-07578-f011:**
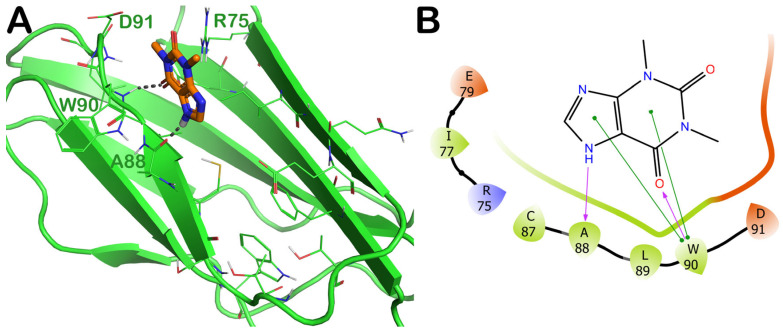
(**A**) Docking output of compound ZINC000018043251 (orange sticks) within the surface of the irisin monomer (PDB ID 4LSD, green cartoon) involved in the dimerization process. The key interacting residues are represented by lines. The H bonds are depicted as dotted lines. Non-polar hydrogen atoms were removed for the sake of clarity. (**B**) A 2D representation of the contacts established by ZINC000018043251 within the irisin monomer interface. Images were prepared using PyMOL and the Ligand-Interaction Diagram application implemented in Maestro.

**Figure 12 ijms-24-07578-f012:**
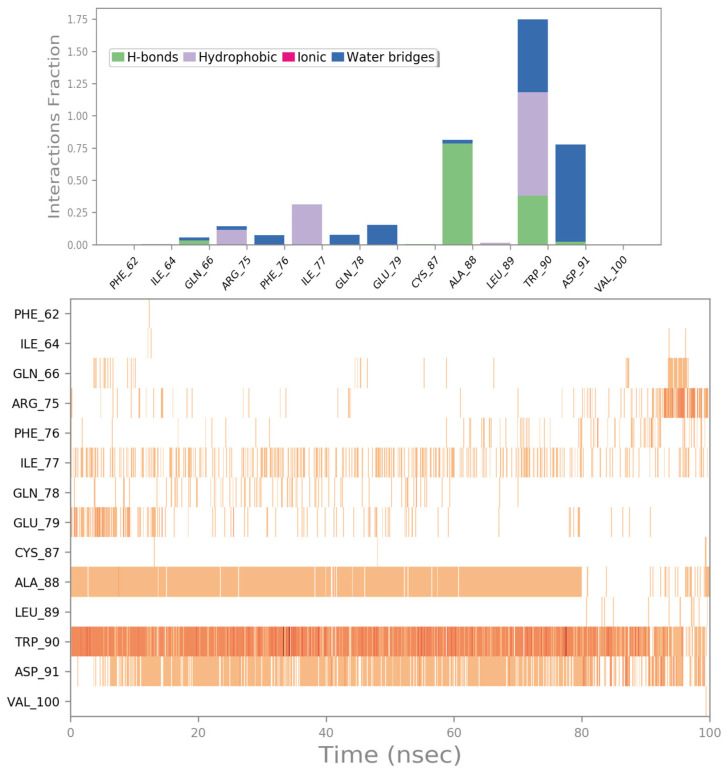
ZINC000018043251 monitored during the MD run. The interactions can be grouped into four types: H bonds (green), hydrophobic (grey), ionic (magenta), and water bridges (blue). The subsequent diagram of the figure illustrates a timeline description of the main interactions. A darker hue of orange indicates that some residues make many distinct contacts with the ligand (Maestro, Schrödinger LLC, release 2020-3).

**Figure 13 ijms-24-07578-f013:**
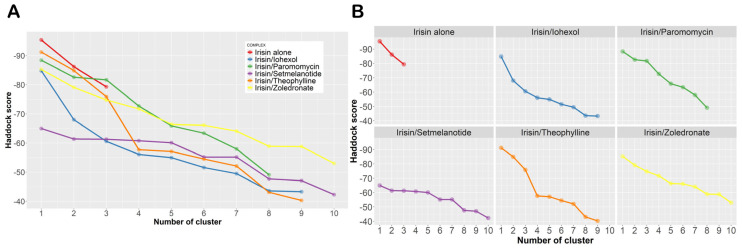
(**A**) Graphical representation of the HADDOCK outputs considering the data reported in Table 4. (**B**) Individual graphs for each studied complex.

**Table 1 ijms-24-07578-t001:** Description of lipodystrophy types, kinds of damage, genes, and proteins involved [8].

Lipodystrophy Type	Mutation/Alteration	Gene Involved	Protein Involved
Congenital generalized lipodystrophy	Recessive mutations	*AGPAT2*	Adipocytes growth (glycerophospholipids and triacylglycerols)
		*BSCL2*	Seipin (endoplasmic reticulum, development of lipid droplets)
		*CAV1*	Caveolin-1 (essential for the normal transport, processing, and storage of fats)
		*PTRF*	Caveolin-3 (caveolar formation)
Familial partial lipodystrophy	Autosomal dominant mutations	*LMNA*	Lamin A/C (stability and strength to cells)
		*PPARG*	PPARγ (key role in lipid and glucose metabolism)
		*PLIN1*	Perilipin-1 (coats lipid storage droplets)
		*AKT2*	AKT Kinase (metabolism, proliferation, growth, and angiogenesis)
	Recessive mutations	*CIDEC*	CIDE-3 (lipolysis restriction and favored storage)
		*LIPE*	Lipase E, Hormone Sensitive (mobilization of triglycerides from adipose tissue)
Acquired lipodystrophies	Autoimmune destruction of adipocytes	*-*	-
Acquired partial lipodystrophy	Mutations	*LMNB*	Lamin-like protein (maintain integrity of nuclear structures in response to mechanical stress)

**Table 2 ijms-24-07578-t002:** Output of the high-throughput docking campaign, with the ranking of approved drugs based on the applied computational protocol.

No	Compound	Status	XP Score (kcal/Mol)	SP Score (kcal/Mol)	ΔG_bind_ (kcal/Mol)	Binding Stability ^a^
1	ZINC000003830946 PID: 3730 Iohexol (Omnipaque)	FDA-approved	−7.70	−5.52	−61.47	yes
2	PID: 11993702 Setmelanotide (Imcivree)	FDA-approved EMA-approved	−6.10	−7.24	−56.87	yes
3	ZINC000008551963 PID: 148197 Diquafosol (Diquas)	Experimental FDA UNII 7828VC80FJ PMDA-approved	−8.17	−5.02	−54.91	yes
4	ZINC000008214585 PID: 3037209 Isepamicin	Experimental FDA UNII G7K224460P PMDA-approved	−7.94	−5.70	−51.72	no
5	ZINC000003830947 PID: 65492 Iopamidol	FDA-approved	−5.94	−5.84	−51.49	no
6	ZINC000060183170 PID: 165580 Paromomycin (Humatin)	FDA-approved	−7.23	−6.18	−50.14	yes
7	ZINC000085504689 PID: 5281771 Echinacoside	Experimental FDA UNII I04O1DT48T	−5.68	−5.36	−48.36	no
8	ZINC000085552699 PID: 5486699 Troxerutin	Experimental FDA UNII 7Y4N11PXO8	−7.45	−5.13	−46.18	no
9	ZINC000003830273 PID: 2327 Benserazide (component of Prolopa)	FDA-approved	−6.71	−6.37	−44.16	no
10	ZINC000238809356 PID: 13943297 Astragaloside IV	Experimental FDA UNII 1J6XA9YCFV	−6.46	−5.09	−43.39	no
11	ZINC000003946372 PID: 9808998 Edotecarin	Experimental FDA UNII 1V8X590XDP	−6.00	−5.91	−43.32	no
12	ZINC000003788703 PID: 444020 Voglibose (Basen)	PMDA-approved Experimental FDA UNII S77P977AG8	−7.46	−5.42	−42.11	yes
13	ZINC000008143568 PID: 10621 Hesperidin	Experimental FDA UNII E750O06Y6O	−6.58	−5.03	−41.69	no
14	ZINC000008214483 PID: 37768 Amikacin (Arikayce)	FDA-approved EMA-approved	−7.11	−5.25	−41.25	no
15	ZINC000049538629 PID: 6451084 Salvianolic acid B	Experimental FDA UNII C1GQ844199	−5.10	−6.04	−40.67	yes
16	ZINC000003977952 PID: 11333 Lactulose (Generlac)	FDA-approved	−6.38	−5.85	−35.66	no
17	ZINC000004096261 PID: 638678 Protirelin (Thyrel TRH)	FDA-approved	−5.63	−5.34	−34.88	no
18	ZINC000018043251 PID: 2153 Theophylline (Quibron-T)	FDA-approved	−5.17	−5.55	−33.25	yes
19	ZINC000003938642 PID: 451417 Thymopentin	Experimental FDA UNII O3Y80ZF13F	−5.32	−5.92	−32.73	no
20	ZINC000003803652 PID: 68740 Zoledronate (Zometa)	FDA-approved EMA-approved	−6.33	−5.53	−32.38	yes
21	ZINC000085537042 PID: 41774 Acarbose (Glucobay; Precose)	FDA-approved	−6.65	−5.11	−31.46	no
22	ZINC000000057624 PID: 439260 Noradrenaline (Levophed)	FDA-approved	−5.03	−5.49	−30.59	no
23	ZINC000002040854 PID: 439224 Carnosine (Sevitin)	Experimental FDA UNII 8HO6PVN24W PMDA-approved	−6.04	−5.25	−29.54	no

PMDA: Medical Devices Agency of Japan; FDA: U.S. Food and Drug Administration; EMA: European Medicines Agency; PID: PubChem ID; ^a^ the binding stability was assessed employing the MD simulation technique.

**Table 3 ijms-24-07578-t003:** Predicted binding free energies (ΔG_bind_) from the MM-GBSA calculations and the energy components of the best-ranked compounds.

Entry	ΔG_vdw_ ^a^ (kcal/mol)	ΔG_coul_ ^b^ (kcal/mol)	ΔG_Hbond_ ^c^ (kcal/mol)	ΔG_Lipo_ ^d^ (kcal/mol)	ΔG_Pack_ ^e^ (kcal/mol)	ΔG_SolGB_ ^f^ (kcal/mol)	ΔG_bind_ ^g^ (kcal/mol)
ZINC000003830946 (Iohexol)	−33.84	−37.12	−4.39	−10.45	−0.11	23.05	−60.71
ZINC000060183170 (Paromomycin)	−32.14	−93.71	−5.52	−12.38	0.00	89.45	−49.13
ZINC000003803652 (Zoledronate)	−11.67	42.11	−3.58	−2.78	−3.09	−31.73	−29.42
PID: 11993702 (Setmelanotide)	−54.67	−113.82	−6.28	−15.23	−5.15	98.12	−62.41
ZINC000018043251 (Theophylline)	−19.56	−18.92	−1.11	−5.19	−7.69	13.18	−35.29

^a^ Contribution of van der Waals interaction energy to the binding free energy; ^b^ Contribution of Coulomb energy to the binding free energy; ^c^ Hydrogen-bonding contributions to the binding free energy; ^d^ lipophilic energy contribution to the binding free energy; ^e^ π−π packing energy contribution to the binding free energy; ^f^ generalized Born electrostatic solvation energy contribution to the binding free energy; ^g^ total binding free energy.

**Table 4 ijms-24-07578-t004:** HADDOCK outputs regarding the protein–protein docking of irisin monomers alone and in the presence of the selected FDA-approved drugs. Clusters are ordered by the Z-scores provided by HADDOCK.

Complex	Clusters									
	1	2	3	4	5	6	7	8	9	10
Irisin alone	−95.4 ± 3.5 (87%)	−86.2 ± 5.4 (8%)	−79.3 ± 2.7 (5%)							
Irisin/iohexol	−84.8 ± 6.5 (31%)	−68.0 ± 4.8 (28%)	60.6 ± 1.0 (11%)	−56.1 ± 5.5 (3%)	−55.0 ± 8.0 (5%)	−51.6 ± 3.9 (7%)	−49.5 ± 2.7 (6%)	−43.6 ± 11.9 (3%)	−43.3 ± 3.1 (3%)	
Irisin/paromomycin	−88.4 ± 7.0 (14%)	−82.6 ± 3.0 (52%)	−81.7 ± 1.4 (11%)	−72.7 ± 11.2 (10%)	−65.9 ± 8.4 (5%)	−63.4 ± 3.2 (3%)	−58.0 ± 8.2 (3%)	−49.1 ± 11.3 (2%)		
Irisin/zoledronate	−85.3 ± 3.1 (41%)	−79.1 ± 5.7 (19%)	−74.8 ± 8.5 (8%)	−71.7 ± 6.3 (5%)	−66.4 ± 7.8 (6%)	−66.1 ± 6.4 (3%)	−64.1 ± 11.6 (4%)	−58.9 ± 8.7 (4%)	−58.8 ± 8.8 (7%)	−53.0 ± 5.0 (3%)
Irisin/setmelanotide	−65.0 ± 3.7 (22%)	−61.4 ± 2.1 (24%)	−61.3 ± 9.2 (3%)	−60.8 ± 2.0 (11%)	−60.1 ± 7.4 (5%)	−55.2 ± 3.3 (14%)	−55.2 ± 9.5 (8%)	−47.7 ± 7.4 (5%)	−47.1 ± 1.4 (4%)	−42.3 ± 3.4 (4%)
Irisin/theophylline	−91.2 ± 3.0 (11%)	−84.9 ± 5.4 (50%)	−75.9 ± 3.1 (9%)	−57.7 ± 4.5 (6%)	−57.1 ± 2.9 (8%)	−54.5 ± 7.1 (3%)	−52.1 ± 4.0 (6%)	−43.1 ± 1.0 (4%)	−40.3 ± 6.9 (3%)	

**Table 5 ijms-24-07578-t005:** The most promising FDA-approved drugs (with related features) potentially able to interfere with irisin dimerization.

Drugs	Marketed Indication	Note
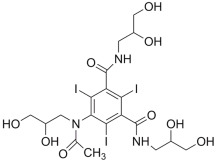 Iohexol	Diagnostic tool used for X-ray imaging	Blocks X-rays as they pass through the body
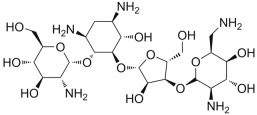 Paromomycin	Treatment of acute and chronic intestinal amebiasis, cutaneous and mucocutaneous leishmaniasis, and as an adjuvant for the management of hepatic coma	Aminoglycoside antibiotic that inhibits protein synthesis by binding to 16S ribosomal RNA
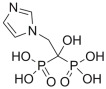 Zoledronic acid	Treatment of malignancy-associated hypercalcemia, multiple myeloma, and bone metastasis from solid tumors	Bisphosphonate; acts on bone tissue by inhibiting the bone resorption process mediated by osteoclasts
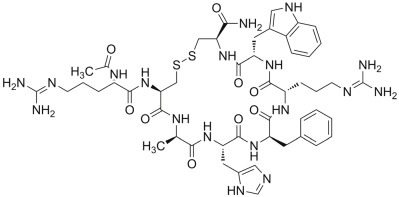 Setmelanotide	Treatment for patients with proopiomelanocortin, proprotein subtilisin/kexin type 1, or leptin deficiencies. Used for treating genetic obesity caused by the mentioned rare single-gene mutations	Melanocortin 4 receptor agonist
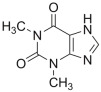 Theophylline	Treatment of the symptoms and reversible airflow obstruction associated with chronic asthma and other chronic lung diseases, such as emphysema and chronic bronchitis	Competitively inhibits type III and type IV phosphodiesterase (PDE)

## Data Availability

Not applicable.

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
