# Peer review of "Disruption of Irisin Dimerization by FDA-Approved Drugs: A Computational Repurposing Approach for the Potential Treatment of Lipodystrophy Syndromes"

_ijms, 2023, doi:10.3390/ijms24087578_

Round 1

Reviewer 1 Report

The manuscript can be accepted after addressing the following remarks:

-The abstract is too general, authors need to provide obtained results such as docking scores, MM-GBSA binding energies and RMSD values.

- In the introduction, authors must provide a brief overview of CADD methods used and and the repurposing approach.

- In results and discussion: authors must specify the number of drugs and the docking scores range for XP and SP filters.

- Based on what authors classified compounds into stable or not in table 1.

- Author must calculate MM-GBSA after MD and compare the results to post docking MM-GBSA results.

- why authors use MacroModel and Ligprep, since the minimization can be obtained using ligprep.

Author Response

Reviewer 1

-The abstract is too general, authors need to provide obtained results such as docking scores, MM-GBSA binding energies and RMSD values.

Authors: thank you for your suggestions. Accordingly, we provided some information about the computational results for the retrieved drugs in the abstract.

- In the introduction, authors must provide a brief overview of CADD methods used and and the repurposing approach.

Authors: according to the reviewer’s comment, we improved the introduction with CADD methods and the repurposing approach.

- In results and discussion: authors must specify the number of drugs and the docking scores range for XP and SP filters.

Authors: we thank the referee for the suggestion. The total number of drugs in the prepared dataset (FDA-approved, world-approved and investigational) is 6,901, while after the screening we found 141 drugs considering SP and 96 drugs considering XP (docking score lower than -5 kcal/mol) for a total of 237 entries to evaluate. These entries were filters following the procedure mentioned in results and discussion “As a result, only compounds able to strongly interacts with the key residues involved in the dimerization (visual inspection) with the unique binding mode were finally selected considering both scoring functions (XP and SP), docking scores lower than -5.00 kcal/mol and a satisfactory ΔGbind”. Details in this sense were added to results and discussion and material and methods sections.

- Based on what authors classified compounds into stable or not in table 1.

Authors: as reported in the main text (row 158-166) at the beginning of paragraph 2.2 Molecular dynamics simulation, we selected only compounds able to maintain the binding to the irisin monomers and it was assessed by visual inspection of the replicated trajectories for each complexes considered. Accordingly, in Table 2 (there was a mistake and the correct numbering for the table is Table 2) we added the column refereed to the binding stability. In order to better understand the classification, we added a footnote on the Table 2

- Author must calculate MM-GBSA after MD and compare the results to post docking MM-GBSA results.

Authors: we thank the referee for the valuable suggestion. Accordingly, we calculated the MM-GBSA on the whole trajectory using the script available in Desmond namely thermal_mmgbsa.py, also reporting the energy components contribution to the binding energy. Results, a brief discussion, including the comparison with the output obtained by molecular docking is reported in the main text (Table 3) and the methodology was described in the materials and methods section.

- why authors use MacroModel and Ligprep, since the minimization can be obtained using ligprep.

Authors: usually, when we treated databases downloaded from website, such as ZINC database, we preferred to use MacroModel for the minimization and above all to avoid possible errors in the structures, during the 3D conversion. In fact, Macromodel is able to assign/correct Lewis structures if the software found wrong ones. This event can occur when the structures are downloaded and Macromodel can avoid these types of errors. After that, Ligprep is used to assign the correct protonation state to the compounds considering the desired pH. We always performed this kind of preparation for the databases from several and several years.

Reviewer 2 Report

The authors present a computational study on identifying FDA-approved drugs that can disrupt irisin dimerization by conducting molecular docking and molecular dynamics simulations. The study's findings are of significant potential in preventing irisin dimerization, and with the suggested revisions, it could make a valuable contribution to the field. However, I believe the manuscript would benefit from some significant revisions as certain inconsistencies and insufficient aspects need to be addressed. I suggest some major revisions as follows:

1: Is there any established binder (ligand) for the receptor that could be used as a reference point?

2. It is recommended that the ligands in Table 1 be ranked according to their DGbind values, as this metric possesses greater physical significance.

3. If the focus is exclusively on the FDA-approved data, it may be unnecessary to display the remaining datasets.

4. It is suggested that an additional column be added to Table 3, displaying the 2D chemical structure of the compounds.

5. It is recommended to contemplate the utilization of MD simulation trajectories for re-calculating the MMGBSA.

6. The rank order of MMGBSA, two Glide dockings, and HADDOCK displays a considerable variance. In Table 1, it can be observed that Zoledronate and Theophylline exhibit much lower affinity towards the receptor in comparison to the other ligands. However, in HADDOCK, the ranks of these ligands differ. Therefore, there is a requirement for the convergence of the results, and further elaboration may be necessary to comprehend these outcomes.

7. Please document the size of each cluster for Table 2.

Author Response

Reviewer 2

The authors present a computational study on identifying FDA-approved drugs that can disrupt irisin dimerization by conducting molecular docking and molecular dynamics simulations. The study's findings are of significant potential in preventing irisin dimerization, and with the suggested revisions, it could make a valuable contribution to the field. However, I believe the manuscript would benefit from some significant revisions as certain inconsistencies and insufficient aspects need to be addressed. I suggest some major revisions as follows:

1: Is there any established binder (ligand) for the receptor that could be used as a reference point?

Authors: unfortunately there is no any previous work highlighting possible irisin dimerization disruptors or ligands. In fact, as stated in the Conclusion section this work is the first to identify possible irisin dimerization disruptors “Notably, this kind of computational screening was carried out for the first time in this paper, providing relevant hints on the possible role of chemical entities in disrupting irisin dimerization.” Accordingly, no established binders are described.

  1. It is recommended that the ligands in Table 1 be ranked according to their DGbind values, as this metric possesses greater physical significance.

Authors: in order to address the point, we reorganized the Table 2 (wrong numbering was detected in the original version) according to the DG values.

  1. If the focus is exclusively on the FDA-approved data, it may be unnecessary to display the remaining datasets.

Authors: considering that, as specified in the materials and methods section, we used not only FDA-approved drugs but also other world-approved and investigational drugs, we considered in the table the relevant output for the whole dataset. Accordingly, we reported in the title FDA-approved since the drugs retrieved that can be considered as disruptors of the irisin dimerization were FDA-approved, but we also highlighted that some investigational drugs can be further assessed for their ability to bind irisin monomer. In summary, we prefer to maintain the current organization of the manuscript, giving some relevance to the other retrieved drugs.

  1. It is suggested that an additional column be added to Table 3, displaying the 2D chemical structure of the compounds.

Authors: thank you for the comments, we added the 2D chemical structures of the compounds in Table 5.

  1. It is recommended to contemplate the utilization of MD simulation trajectories for re-calculating the MMGBSA.

Authors: we thank the referee for the valuable suggestion. Accordingly, we calculated the MM-GBSA on the whole trajectory using the script available in Desmond namely thermal_mmgbsa.py, also reporting the energy components contribution to the binding energy. Results, a brief discussion, including the comparison with the output obtained by molecular docking is reported in the main text (Table 3) and the methodology was described in the materials and methods section.

  1. The rank order of MMGBSA, two Glide dockings, and HADDOCK displays a considerable variance. In Table 1, it can be observed that Zoledronate and Theophylline exhibit much lower affinity towards the receptor in comparison to the other ligands. However, in HADDOCK, the ranks of these ligands differ. Therefore, there is a requirement for the convergence of the results, and further elaboration may be necessary to comprehend these outcomes.

Authors: we’d like to point out that different methods were used to rank the compounds, providing a more reliable and robust selection of the drugs. In fact, considering the docking scores, they were applied as the filters for retrieving compounds with unique binding modes considering the two scoring functions (SP and XP). After that we assessed, using MMGBSA, if the retrieved poses showed satisfactory DG binding. Accordingly and how it was well established, the correlation of docking score with MMGBSA is not particularly good and for this reason coupling the two methods can improve the reliability of the work. Different the case of the haddock. In fact, haddock did not measure the affinity of the compound to the irisin monomer, but the affinity of the irisin monomer to the other monomer to form the dimer. So, including the compounds in the calculation is indicative that when a selected drug is present, the irisin dimer can be formed with higher difficult considering the huge number of clusters found and the decrease of affinity found. So, it is not possible to compare the energies found by docking/MMGBSA and haddock since they are referred to a different situation (ligand-protein docking for XP/SP/MMGBSA and protein-protein docking for haddock).

  1. Please document the size of each cluster for Table 2.

Authors: we inserted the percentage of docked solution for each cluster in the Table 4. Notably, the clusters are ordered by Z-score provided by haddock and no by the number of haddock docking solutions.

Reviewer 3 Report

The manuscript describes a computational protocol to identify FDA-approved drugs that can interfere with irisin dimerization and could be useful as novel therapeutic options in LD. The work presents a modern approach to the in silico experiment in an interesting and concise way. The computational approach to the subject and the preparation of the drug database is very professional, the database itself refers to the latest FED data. It is valuable that a detailed description of the procedures is described. I believe that the manuscript should be published as soon as possible.

Author Response

Reviewer 3

The manuscript describes a computational protocol to identify FDA-approved drugs that can interfere with irisin dimerization and could be useful as novel therapeutic options in LD. The work presents a modern approach to the in silico experiment in an interesting and concise way. The computational approach to the subject and the preparation of the drug database is very professional, the database itself refers to the latest FED data. It is valuable that a detailed description of the procedures is described. I believe that the manuscript should be published as soon as possible.

Authors: thank you for the positive evaluation of the paper and for appreciating our work.

Round 2

Reviewer 1 Report

Authors addressed the points in as satisfactory manner and I recommend it for publication.

Reviewer 2 Report

Accept in present form.